# Success Metrics for Hepatitis C Elimination Among People Who Inject Drugs: A Scoping Review of Indicators in Harm Reduction

**DOI:** 10.3390/ijerph22071036

**Published:** 2025-06-28

**Authors:** David S. Kremer, Pauline Elizabeth Gatmaytan, Michelle Amanda Rübel, Antoine Flahault, Jennifer Hasselgard-Rowe

**Affiliations:** 1Global Studies Institute, University of Geneva, 1205 Geneva, Switzerland; pauline.gatmaytan@etu.unige.ch (P.E.G.); michelle.ruebel@etu.unige.ch (M.A.R.); 2Faculty of Medicine, Institute of Global Health, University of Geneva, 1211 Geneva, Switzerland; antoine.flahault@unige.ch (A.F.); jennifer.hasselgard-rowe@unige.ch (J.H.-R.)

**Keywords:** harm reduction, People Who Inject Drugs (PWID), Hepatitis C Virus (HCV), scoping review, hepatitis C indicators, hepatitis C elimination

## Abstract

This study aimed to identify and synthesize the success metrics used to assess hepatitis C elimination among people who inject drugs (PWID) through harm reduction strategies. A scoping review was performed by searching across three databases to identify systematic reviews that discussed hepatitis C in PWID within the context of harm reduction. The studies were then analyzed for success metrics used to describe hepatitis C in PWID. The indicators used were prevalence, incidence, screening, treatment uptake, treatment completion, and sustained virologic response. A total of fourteen systematic reviews were included. The most frequently reported indicators were prevalence and incidence, addressed in eight/seven systematic reviews, respectively. In contrast, screening, treatment uptake, and treatment completion were less commonly reported, with only two reviews addressing screening and treatment uptake, and a single review reporting treatment completion. Similarly, sustained virologic response (SVR) was reported in only two systematic reviews. Seven additional indicators were reported. Prevalence and incidence are the dominantly used HCV indicators, while others are often neglected. Inconsistencies in measurements and reporting can be found for all indicators. This study reports a gap regarding indicators beyond prevalence and incidence, inconsistent measurement approaches, and a lack of standardized frameworks.

## 1. Introduction

Hepatitis C is a liver inflammation caused by the hepatitis C virus (HCV), which can lead to both acute and chronic infections, with the latter potentially being fatal. It is primarily transmitted through blood-to-blood contact, such as sharing needles or syringes [1]. Therefore, people who inject drugs (PWID) are an important population at risk [2] and will play a major role in future HCV control [3].

Most acute infections remain asymptomatic, but around 70% of cases will develop a chronic infection. While there is currently no vaccine available against HCV, individuals with chronic infections rely on proper treatment to live healthy lives and prevent further transmission, though reinfection is still possible [1]. Modern direct-acting antiviral medicines (DAAs) allow not just treatment but cure of the infection to be the primary goal of therapy [4], and more than 95% of infected individuals can be cured [1]. The cost of DAAs has significantly decreased with the availability of generics, leading to treatment costs of around USD 60 per 12-week course in many countries [5].

Globally, PWID are at a significantly higher risk of acquiring HCV, with an estimated 39% of them currently living with the virus. In settings without effective harm reduction strategies, the prevalence of HCV among this population remains a substantial public health challenge. Harm reduction is a public health strategy designed to mitigate the risks associated with drug use by focusing on improving health and social outcomes instead of criminalization and punitive measures. Among PWID, harm reduction programs serve as a cornerstone for addressing the challenges of infectious diseases, including HCV, which remains a critical public health concern [6].

The goal of global HCV elimination by 2030, as outlined in the United Nations’ Sustainable Development Goals (SDGs), requires comprehensive harm reduction strategies [7]. Despite the proven benefits of harm reduction, there is a need to evaluate the specific indicators that best reflect progress in HCV elimination. The viral hepatitis elimination target by 2030 as part of the SDGs calls for not only reducing HCV incidence but also improving the accessibility of antiviral treatments, particularly for PWID, who face unique barriers to care [7]. This scoping review aims to examine the indicators used to measure success in HCV elimination within harm reduction programs. The relevant literature was reviewed to identify existing success metrics, including prevalence, incidence, screenings, and treatment uptake among PWID. While metrics such as prevalence and incidence are often used to assess the scale of the epidemic, other indicators may be critical for understanding the effectiveness of harm reduction strategies in improving health outcomes.

This review highlights a gap in the current literature and suggests that a variety of indicators may offer valuable insights into the factors influencing HCV transmission and treatment outcomes among PWID. By assessing current metrics and how they are used, this study aims to provide a better understanding of metrics and inform future research, policy, and program development.

The guiding research questions for this review are as follows:(1)What are the current success metrics for HCV in harm reduction programs targeting PWID?(2)How do these metrics reflect outcomes related to HCV prevalence, incidence, screening, treatment uptake, completion, and SVR?

This review concludes with recommendations for addressing the gap in existing research and improving the global response to HCV in PWID populations, ultimately supporting the achievement of the 2030 elimination goal.

## 2. Materials and Methods

This scoping review follows the five-stage framework provided by Arksey and O’Malley [8] and the recommendations by Pollock et al. [9], focusing on exploring HCV indicators in harm reduction among PWID.

The selection of HCV indicators was informed by an initial screening of the comprehensive 2023 report on “HIV, Hepatitis & Drug Policy Reform” published by the Global Commission on Drug Policy. Following a critical review of the report and its referenced literature, the following key questions were developed to guide the present analysis: What are the current success metrics for HCV in harm reduction? How are these metrics measured over time, and how do they reflect the outcomes related to HCV prevalence, incidence, screening, and treatment uptake?

The identified HCV indicators were prevalence, incidence, screening, treatment uptake, treatment completion, and sustained virological response as potential success metrics for elimination through harm reduction.

To identify studies relevant to the research question, the existing literature was assessed in three databases—PubMed, Web of Science, and EmBase. The search formula used across the databases included the following keywords to ensure consistency among the databases, given their respective formatting:*Harm Reduction**People who Inject Drugs or People who Use Drugs**Hepatitis C Prevalence or HCV Prevalence**Hepatitis C Incidence or HCV Incidence**Hepatitis C Treatment Uptake or HCV Treatment Uptake**Hepatitis C Treatment Completion or HCV Treatment Completion**Hepatitis C Screening or HCV Screening**Sustained Virological Response*

The exact search query can be found in Appendix A (Figure A1). Mesh terms were excluded during the search to ensure the specificity of the results. The publishing period of potential records was limited from 2015 to 2024 to align with the Sustainable Development Goals established in 2015.

The exclusion criteria were as follows: studies that are not HCV-specific, studies that do not discuss the selected HCV indicators, and studies that did not report on PWID. Non-English publications and inaccessible reports were also excluded. Due to the high number of search results, an additional final exclusion criterion was added to exclude studies that were not systematic reviews. Systematic reviews were included as these reports are considered the highest quality of evidence available. After the selection process, a manual screening was conducted. The selection process is presented in Figure 1.

Data from the selected studies were extracted by the authors individually and cross-checked by a second author. This extraction process was performed in two rounds: the first identified the indicators and the second examined how they were measured.

The extracted data include several study characteristics (e.g., author(s), publication year, publication journal, country/region, study design, and number of included articles the studies used) (Table 1).

## 3. Results

A total of 495 records were identified across the three databases: PubMed had 136 results, Web of Science had 135 results, and EmBase had 224 results. The total was reduced to 245 results after automatically removing duplicates in EndNote 21 and a manual follow-up screening for duplicates. This was further reduced to 16 upon limiting study types to systematic reviews. Three additional records were identified through the reference lists of obtained articles. After a final manual check, 14 studies were included in this review, as shown in Figure 1 [10,11,12,13,14,15,16,17,18,19,20,21,22,23].

### 3.1. Study Characteristics

Table 1 displays the study characteristics, including the number of included studies, study types, and regional focus. The number of included studies providing data in the systematic reviews ranged from 3 to 142, with the median being 34.5 studies. Most reviews had very broad or no limitations regarding the type of included studies, with only two reviews being restricted to a single study type (cross-sectional, systematic reviews). The majority of reviews did not have a regional restriction apart from four reviews which focused on data from a single country (Iran, Afghanistan, Vietnam, Iran) while two focused on more than one country (Lebanon, Palestine, and Syria; EU and comparable countries).

### 3.2. Indicator Frequencies

The most frequently reported indicators were prevalence and incidence, addressed in eight and seven systematic reviews, respectively. In contrast, screening, treatment uptake, and treatment completion were less commonly reported, with only two systematic reviews addressing screening and treatment uptake, and a single review reporting on treatment completion. Similarly, SVR, a critical outcome of HCV treatment, was reported in only two systematic reviews.

Additionally, seven other indicators not identified within this list of success metrics were reported, including genotype diversity, risk behavior, transmission, visit, impact of exposure, treatment adherence, and linkage to care (Table 2).

### 3.3. Indicator Details

A detailed breakdown of indicator reporting and measurement is presented in Table A1.

#### 3.3.1. Prevalence

Prevalence of HCV infection was commonly reported as the percentage of infected individuals among the population of interest. As part of their meta-analysis, five reviews reported a pooled estimate of prevalence [12,14,15,17,21] while two others [11,13] reported a range of prevalence without a pooled estimate. Finally, one review only reported alterations of prevalence in the form of odds ratios or risk ratios [23].

Two reviews reported prevalence solely based on serological tests for HCV antibodies, which was referred to as the prevalence of previous infections [12,21], while four reviews additionally reported the prevalence of current infections, based on reverse transcriptase Polymerase Chain Reaction (rt-PCR) testing for the viral RNA (ribonucleic acid) [13,14,15,17]. One review did not specify the origin of the reported prevalence data but simply stated to have found “large difference[s] in baseline prevalence of [...] HCV” [11]. Lastly, one report chose an umbrella review method and therefore did not include details about the testing criteria in the primary studies [23].

Moreover, one review mentioned the eligibility of studies that established prevalence data based on self-reporting but did not include any studies of that type [14]. Two reviews also reported trends of prevalence over time [14,21].

#### 3.3.2. Incidence

Incidence of HCV infection was commonly reported as the number of new infections during 100 person-years among the population of interest, with three reviews additionally reporting pooled estimates as part of their meta-analysis [10,16,18]. However, one of those only reported pooled estimates for male versus female PWID, not for PWID overall [16], and another reported reinfection incidence only [18].

Two reviews reported a range of HCV incidence without a pooled estimate [11,20], whereas one review only included a single study providing an incidence rate [14]. Finally, one review only reported alterations of incidence in the form of odds ratios or risk ratios [23].

Three reviews included studies that used either RNA or serological testing to establish an incidence rate [10,16,20]. Focusing on reinfections, one review included studies that either used HCV RNA testing after SVR or detection of a different HCV genotype through viral sequencing [18].

The other three reviews remain less clarified: Arum et al. [11] used follow-up testing as an inclusion criterion but did not provide further detail. Chemaitelly et al. [14] cited both RNA and serological test-based incidence data being extracted, but did not specify which test was used in the single incidence study they included in their review. Tonin et al. [23] chose an umbrella review method and therefore did not include details about the testing criteria in the primary studies.

#### 3.3.3. Screening

Representing the first step of linkage to care, the percentage of PWID being screened for HCV infection was only reported in one review. More specifically, even within this review, only one study reported on the outcome of “conducting baseline HCV Evaluation within 3 months” [22].

#### 3.3.4. Treatment Uptake

Treatment uptake was reported based on the percentage of individuals who started HCV treatment among PWID in two reviews. One of those included studies reported receiving prescriptions for at least one DAA (directly acting antiviral) within 8 weeks, 180 days, or 1 year, respectively, while most included studies did not specify the time frame [22]. The other review did not specify the time frame either [19].

#### 3.3.5. Treatment Completion

The percentage of PWID who completed treatment was reported in one of the reviews included [22]. Within the included studies, the definition of completion ranged from 8 to 12 weeks of DAA treatment as well as some requiring a follow-up after 12–24 weeks, while for most studies, it was not clarified.

#### 3.3.6. Sustained Virological Response

Usually measured through viral load in the blood, sustained virological response (SVR) after completion of antiviral treatment is the goal of HCV therapy in individuals and was reported in two reviews [19,22]. While Schwarz et al. [22] included studies that reported either 12- or 24-week SVR as a marker of successful treatment, Oru et al. [19] only referred to SVR12.

## 4. Discussion

To the best of our knowledge, this scoping review is the first study to examine HCV indicators in order to guide the monitoring and evaluation of harm reduction programs for PWID. The key strength of this article is acknowledging an evidence gap for the measurement, standardization, and potential application of success indicators in HCV elimination strategies. The indicators of interest for this review were prevalence, incidence, screening, treatment uptake, treatment completion, and sustained virological response to map the cascade of care for HCV. The included studies showed a limited focus on indicators beyond prevalence and incidence, inconsistent measurement approaches, and a lack of standardized frameworks for measuring indicators such as treatment uptake, treatment completion, screening, and sustained virological response. This finding aligns with previous research, which described a lack of standardization in interventions and significant heterogeneity in systematic reviews [23]. Exploring the utilization of these indicators and analyzing how they can be measured is beneficial for both research and building effective strategies to eliminate HCV in PWID.

While prevalence was the most commonly reported indicator in the included reviews, it becomes obvious that current evidence is insufficient to judge significant progress toward elimination. While half of the reviews report both the prevalence of previous and current infections, the other half report either only previous infections, only alterations of prevalence, or do not specify the prevalence altogether. This variation may be due to the underlying primary data yet still hinders direct comparison.

This research reveals links among relevant indicators that should be further examined. Several reviews refer to indicators of interest, but these were not comprehensively measured as a success metric. For example, SVR is used as a reference for incidence rate, treatment uptake, and treatment completion but is not reported as a success indicator by itself [18], although it is an important surrogate for treatment completion [22]. Moreover, the few reviews that report SVR as an indicator inconsistently resort to either SVR12 or SVR24, making results harder to translate [19,22].

Specifically, within treatment indicators, there were great inconsistencies between the reviews. Time frames for treatment uptake and completion varied, but more importantly, the lack of detailed information impairs comparability. Nonetheless, even the commonly reported indicators showed a lack of standardization, with different terms being used and some reviews not specifying which tests support their incidence data [11,14,23].

Terminology again highlights the significance of identifying and standardizing success metrics. For example, while referring to measures of incidence, studies used not just the term incidence, but infection rates [18], acquisition risk [11,20], transmission risk [20,23], and notably reinfection rates [18]. While the first three terminologies are potentially confusing, reinfection rates are medically distinct and need to be considered separately to avoid misunderstandings.

Beyond the curated list, the following indicators were also marked as potentially valuable: genotype diversity, risk behavior, transmission, visit, impact of exposure, treatment adherence, and linkage to care.

Future research should explore both frequently used and alternative indicators while striving to establish a standardized framework and terminology that researchers, public health officials, policymakers, healthcare providers, Non-Governmental Organizations, and patients could refer to. Doing so would provide consistency in HCV monitoring and evaluation across multiple regions and ultimately contribute to the global goal of HCV elimination.

## 5. Limitations

The following study limitations should be acknowledged.

Analysis was performed on the selected systematic reviews and not the primary sources on which they are based. Primary research lay outside the scope of this review, which focuses on the available, highest level of evidence. The included publications used various data collection methods and program implementations across different regions, making direct comparisons and generalizability difficult. The reported data may not capture informal practices, introducing the possibility of further gaps. Furthermore, measurements of accessibility (e.g., available facilities, program size, affordability) lie beyond the scope of this review, although their role within harm reduction strategies should not be neglected.

This scoping review relied on data limited to publications in English found on PubMed, Web of Science, and Embase, potentially excluding relevant studies that can be found in other languages and databases.

Finally, the article by Tonin et al. [23] is a systematic review of systematic reviews, also called an umbrella review. It includes one of the other articles used for this study [20]. Because of the specific scope of this study, duplicate publication bias was deemed of low significance in this instance.

## 6. Conclusions

This scoping review analyzes HCV indicators and how these are measured within people who inject drugs. Focusing on prevalence, incidence, screening, treatment uptake, treatment completion, and sustained virological response, a gap is revealed in the literature. This examination calls for a comprehensive and standardized measurement of success indicators, which may have implications for monitoring and evaluating harm reduction programs on global and local levels. While prevalence and incidence are prominent across the literature, they fall short of consistency. This inconsistency also applies to the other selected indicators, which tend to be underreported. This leads to a gap in evidence for the latter steps of the HCV cascade of care, such as repeated screening, treatment uptake, treatment completion, and outcomes like SVR. This disparity suggests a need for more comprehensive systematic reviews that address these critical aspects of HCV management in PWID.

This review underlines the importance of creating a standardized framework to measure and report HCV indicators to support global HCV elimination efforts. Furthermore, upon encountering alternative indicators such as genotype diversity, risk behaviors, transmission patterns, treatment adherence, and linkage to care, researchers should continue to explore the potential of alternative indicators and their application.

Nevertheless, the primary recommendation is for researchers and policymakers to prioritize and establish a uniform framework and definition for HCV indicators. This would allow cross-regional application while assessing the success of harm reduction strategies across the globe. Standardizing this framework would not only enhance the quality of future research but also strengthen global efforts to eliminate HCV, ultimately advocating for the public health of people who inject drugs and beyond.

## Figures and Tables

**Figure 1 ijerph-22-01036-f001:**
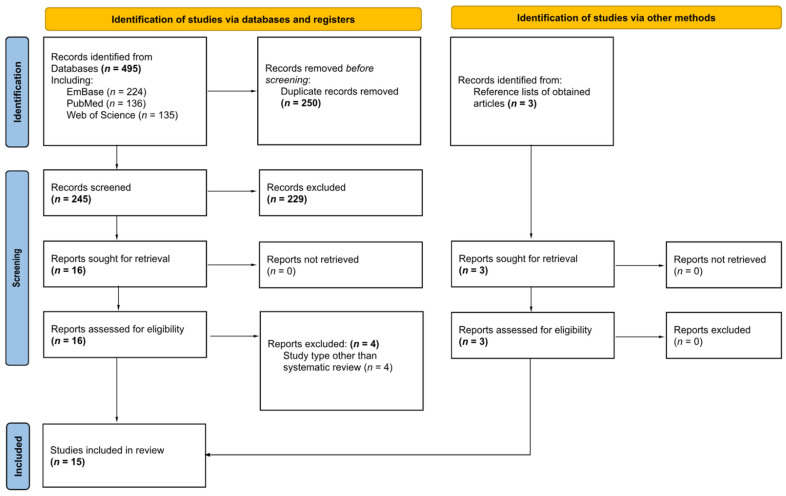
Prisma flowchart of study identification. *n* = number of records. Numbers of records for main steps are in bold font, subcategories and not-applicable steps are in regular font.

**Table 1 ijerph-22-01036-t001:** Study characteristics of included reviews.

Author and Year [Reference Number]	Studies Included	Included Study Types	Country/Region (*n* = Number of Studies)
Artenie, 2023 [10]	65	No study design restrictions.	Global: European (*n* = 26); Western Pacific (*n* = 13); The Americas (*n* = 21); Eastern Mediterranean (*n* = 2); South-East Asia (*n* = 4). (One study included data for Western Pacific and South-East Asia.)
Arum, 2021 [11]	28	No study design restrictions.	Global: North America, Europe, Australia, East Africa, Asia
Behzadifar, 2020 [12]	42	Only cross-sectional studies.	Iran
Chemaitelly, 2015 [13]	3	Any document reporting measures of HCV incidence and/or prevalence based on primary data. Case reports, case series, editorials, letters to editors, and commentaries are excluded.	Lebanon, Palestine, and Syria (all *n* = 1)
Chemaitelly, 2015—Afghanistan [14]	15	Any document reporting serological measures for HCV incidence and/or prevalence based on primary data. Case reports, case series, editorials, letters to the editor, and commentaries are excluded.	Afghanistan
Degenhardt, 2023 [15]	80	Cohort studies without baseline data, case-control studies, and non-original works (e.g., reviews or editorials) were excluded.	Global: studies from 116 countries included
Esmaeili, 2017 [16]	28	Longitudinal studies, including intervention studies.	Global: Australia, Canada, China, Europe, USA
Flower, 2022 [17]	72	Prospective and retrospective studies, observational studies, and seroprevalence data sources. Surveys and screening studies.	Vietnam
Hajarizadeh, 2020 [18]	36	Prospective and retrospective studies, observational cohort studies.	Multi-country: Canada (*n* = 9), USA (*n* = 4), UK (*n* = 4), Spain (*n* = 4), Norway (*n* = 3), Germany (*n* = 2), Multi-country (*n* = 2), Australia, Austria, Belgium, Denmark, Georgia, Greece, Netherlands, Switzerland (all *n* = 1)
Our, 2021 [19]	142	Randomized controlled trials, non-randomized studies, and observational studies.	Multi-country: 34 countries—USA (*n* = 41), Australia (*n* = 27), UK (*n* = 18), Canada (*n* = 17), Turkey, India, China, Kenya, Georgia, Romania, Iran, Ukraine, Myanmar, Mozambique, Pakistan, Egypt, Indonesia, Cameroon, Cambodia, Brazil (all *n* = 1). Complete list of countries not provided, 23 studies unaccounted for.
Platt, 2017 [20]	28	Observational (prospective and retrospective cohorts, cross-sectional surveys, and case-control studies) or experimental studies.	Global: North America (*n* = 13), United Kingdom (*n* = 5), Australia (*n* = 5), Europe (*n* = 4), China (*n* = 1).
Rajabi, 2021 [21]	62	Scientific documents reporting original data (i.e., gathered directly by conducting surveys and laboratory tests on specimens), in the form of a peer-reviewed manuscript, progress report, abstract, technical report, or substantive scientific commentary and reported on epidemiological data.	Iran
Schwarz, 2022 [22]	14	Only comparative studies.	Restriction: EU/EEA/EFTA member states, candidate countries to the EU, and comparable countries such as Australia, Canada, New Zealand, United Kingdom, USA were included. USA (*n* = 6); Canada (*n* = 2); Australia/New Zealand, United Kingdom, Germany, Austria, Italy (all *n* = 1) and one multicenter trial at sites in USA and Europe
Tonin, 2024 [23]	33	Systematic reviews (with or without meta-analysis) that included primary studies of any design (interventional, observational).	Multi-country (no further details due to study design)
Esmaeili, 2017 [16]	28	Longitudinal studies, including intervention studies.	Global: Australia, Canada, China, Europe, USA

**Table 2 ijerph-22-01036-t002:** Indicator frequency. The numbers of included reviews that reported on indicators of interest are shown in blue. Other indicators are shown in yellow and summarized in orange.

Indicators	Frequency	Author and Year
Prevalence	8	Artenie, 2023 [10], Arum, 2021 [11], Behzadifar, 2020 [12], Chemaitelly, 2015 [13], Chemaitelly, 2015 Afghanistan [14], Degenhardt, 2023 [15], Flower, 2022 [17], Rajabi, 2021 [21]
Incidence	7	Arum, 2021 [11], Chemaitelly, 2015 Afghanistan [14], Esmaeili, 2017 [16], Hajarizadeh, 2020 [18], Platt, 2017 [20]
Screening	2	Oru, 2021 [19]
Treatment Uptake	2	Schwarz, 2022 [22], Oru, 2021 [19]
Treatment Comp.	1	Schwarz, 2022 [22]
SVR	2	Oru, 2021 [19], Schwarz, 2022 [22]
Genotype Diversity	1	Chemaitelly, 2015 [13]
Risk Behavior	1	Tonin, 2024 [23]
Transmission	1	Platt, 2017 [20], Tonin, 2024 [23]
Visit	1	Schwarz, 2022 [22]
Impact of Exposure	1	Platt, 2017 [20]
Treatment Adherence	1	Schwarz, 2022 [22]
Linkage to Care	1	Oru, 2021 [19]
Others	7	Oru, 2021 [19], Chemaitelly, 2015 [13], Platt, 2017 [20], Tonin, 2024 [23], Schwarz, 2022 [22]

## Data Availability

No new data were created or analyzed in this study.

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
