# Peer review of "Success Metrics for Hepatitis C Elimination Among People Who Inject Drugs: A Scoping Review of Indicators in Harm Reduction"

_ijerph, 2025, doi:10.3390/ijerph22071036_

Round 1

Reviewer 1 Report

Comments and Suggestions for Authors

Kremer et al. presented a review analysis of 14 reviews on the topic of success metrics of HCV elimination in PWID. In the era of DAA, HCV in PWID remains a significant public health concern. It is a simple statistics of success metrics and the conclusion is to some extent a predictable fact. I have no concern on its performance. Instead, multiple concerns come from its presentation.

  1. In the introduction, the study clearly stated two goals (line 72-76). The second goal stated, “How are these metrics measured over time, and how effectively do they reflect outcomes related to HCV prevalence, incidence, screening, and treatment uptake?”. In the sections of Results, Discussion, and Conclusions, I did not find the evidence that the authors achieved this goal. If you could not reach this goal, it should be eliminated.
  2. Table 2 (line 160-164). Instead of colored mark, I suggest that the authors adding one more column to indicate “indicators of interest” and “other indicators”. This may give readers easy reading and understanding.
  3. Issues on HCV virology terms. HCV is an RNA virus so that it requires reverse transcription PCR rather than PCR for detection (line 174). | HCV antibodies (line 172): It is better to use “HCV serological tests”. | HCV antigen (ribonucleic acid, RNA) (line 175): HCV RNA (antigen usually means protein) and you do not need to spell out “RNA”. | “used either RNA- or antibody-testing to establish” (line 194, line 200): HCV RNA or serological tests to establish. | SVR definition (footnote on page 12): SVR12 is defined as undetectable levels of HCV RNA in the blood for 12 weeks after completing antiviral treatment.
  4. Some sentences are strange (ChatGPT?). For instance, HCV success metrics (line 61): I know what the authors mean but this causes confusion. It’s better to state “HCV elimination metrics”.
  5. Also, “mention”: (line 198): mentioned (there are multiples like this throughout the text).
  6. Reinfection in PWID (line 262-263): Reinfection is a salient feature in PWID. The authors made a good point. A little bit of extension regarding reinfection in the metric of incidence is expected.
  7. line 176: difference.

Reviewer 2 Report

Comments and Suggestions for Authors

In an odd sort of way, hepatitis C virus infection is a neglected disease. Its acute infection stage is very often asymptomatic. Viral flares among the approximately 70% of those with chronic infection are Intermittent and rarely life-threatening. End stage manifestations of cirrhosis and hepatocellular carcinoma occur decades after initial infection. Efforts to eliminate HCV, made possible by the development of increasingly effective directly active antiviral agents, are far from meeting international goals. Finally, the preponderance of infection in stigmatized populations of people who inject drugs reduces the perceived social value of prioritizing expanding access to harm reduction and curative treatment needed for elimination. In this context, this scoping review of systematic reviews to identify metrics for assessing HCV disease and its elimination is a useful guide. The lack of consistent metrics for even the most common epidemiological measures – prevalence, incidence, and treatment rates – provides a clear reminder of the extent to which HCV remains a neglected disease.

The writing is clear and sufficiently succinct. Appendix B, with greater detail on the appearance of each metric in the systematic reviews is a useful addition. Several suggestions for improving this manuscript:

  1. Hepatitis should not be capitalized unless it is at the start of a sentence.
  2. This sentence in the abstract, “Prevalence and incidence are prominent HCV indicators. Screening, Treatment Uptake, Completion, and SVR were used less frequently.” is unnecessarily redundant and should be deleted.
  3. At lines 174-5, I am confused by the sentence on testing for current infection. Are there two testing methods (nucleic acid amplification of viral RNA and detection of viral antigens) or just one -- PCR amplification of viral RNA? Clarification is needed.
  4. There is one major limitation that goes unmentioned. Systematic reviews summarize studies of primary research reports and often provide scant methodological detail. The authors do not appear to have reviewed the original papers covered in the systematic reviews. These papers may actually contain more details of many of the six major and at least seven minor metrics that the authors have considered.

Round 2

Reviewer 1 Report

Comments and Suggestions for Authors

None

Comments on the Quality of English Language

None